# Reviews and syntheses: Ice Acidification, the effects of ocean acidification on sea ice microbial communities

Andrew McMinn[1]

[1]Institute of Marine and Antarctic Science, University of Tasmania, Hobart 7001, Tasmania, Australia

*Correspondence to*: Andrew McMinn (andrew.mcminn@utas.edu.au)

**Abstract.** Sea ice algae, like some coastal and estuarine phytoplankton, are naturally exposed to a wider range of pH and $CO_2$ concentrations than those in open marine seas. While climate change and ocean acidification (OA) will impact pelagic communities, their effects on sea ice microbial communities remains unclear.

Sea ice contains several distinct microbial communities, which are exposed to differing environmental conditions depending on their depth within the ice. Bottom communities mostly experience relatively benign bulk ocean properties, while interior brine and surface (infiltration) communities experience much greater extremes.

Most OA studies have examined the impacts on single sea ice algae species in culture. Although some studies examined the effects of OA alone, most examined the effects of OA and either light, nutrients or temperature. With few exceptions, increased $CO_2$ concentration caused either no change or an increase in growth and/or photosynthesis. *In situ* studies of brine and surface algae also demonstrated a wide tolerance to increased and decreased pH and showed increased growth at higher $CO_2$ concentrations. The short time period of most experiments (<10 days) together with limited genetic diversity (i.e. use of only a single strain), however, has been identified as a limitation to the broader interpretation of results.

While there have been few studies on the effects of OA on the growth of marine bacterial communities in general, impacts appear to be minimal. In sea ice also, the few reports available suggest no negative impacts on bacterial growth or community richness.

Sea ice ecosystems are ephemeral, melting and re-forming each year. Thus, for some part of each year organisms inhabiting the ice must also survive outside of the ice, either as part of the phytoplankton or as resting spores on the bottom. During these times, they will be exposed to the full range of co-stressors that pelagic organisms experience. Their ability to continue to make a major contribution to sea ice productivity will depend not only on their ability to survive in the ice but also on their ability to survive the increasing seawater temperatures, changing distribution of nutrients and declining pH forecast for the water column over the next centuries.

## 1 Introduction

Sea ice is widely recognized as one of the most extreme habitable environments on earth (Thomas and Dieckmann, 2003, Martin and McMinn 2017). Organisms living within it can be exposed to temperatures of below -20°C and salinities greater than 200 for extended periods of time (Arrigo, 2014; Thomas and Dieckmann, 2003). These environments also endure long periods of darkness and extremes in nutrient concentration, dissolved gases ($O_2$, $CO_2$) and pH (Thomas and Dieckmann 2003 McMinn et al., 2014). In spite of these conditions, some sea ice habitats are very productive and often support a dense microbial biomass (Arrigo, 2014). They also play a key supporting role in maintaining energy flow through polar marine food webs in winter and spring when sea ice covers much of the ocean surface and there is very little pelagic primary production (Arrigo and Thomas, 2004).

During the annual cycle of sea ice formation and melting, associated organisms are naturally exposed to a large range in gas concentrations, including $CO_2$. Gases are expelled from the forming ice in autumn and winter and become concentrated in the trapped brine (Fig. 1). Consequently, there is a relationship between the $CO_2$ concentration of the brine and that of the original sea water. As dissolved $CO_2$ concentrations continue to rise with ocean acidification (OA), there will also be a commensurate increase in the $CO_2$ concentration in the trapped brine. Macronutrients are also expelled from growing ice and are also concentrated in the brine in autumn but taken up for cell growth in subsequent months (Fig. 1). This review will examine the effects of increasing $CO_2$ concentrations and decreasing pH on sea ice microbial communities. The review focuses on Antarctic sea ice communities as few Arctic studies have been identified.

## 2. Natural pH fluctuations

There are strong seasonal cycles in pH in the seasonal ice-covered waters of the Southern Ocean due to brine drainage from forming sea ice in winter and high levels of biological activity in summer. (Miller et al., 2011). In Prydz Bay and the Ross Sea this has led to seasonal differences of up to 0.6 of a pH unit (Gibson and Trull, 1999; McNeil et al., 2010), which is greater than the 0.4 increase usually predicted for the end of the century (IPCC, 2014). Large seasonal differences in pH have also been observed in McMurdo Sound, Antarctica (Kapsenberg et al., 2015; Matson et al., 2014), where pH was relatively high but stable during winter but was lower and dynamic in summer. Large diurnal fluctuations in pH beneath the sea ice were also observed and it was suggested that these changes could be almost entirely explained by the biological processes of photosynthesis and respiration (Matson et al., 2014).

During the early stages of ice formation in early winter, the concentration of $CO_2$ within the ice is determined solely by physical processes associated with freezing and brine rejection. While low temperatures increase the solubility of $CO_2$, increasing brine salinity during sea ice growth leads to super-saturation (Papadimitriou et al., 2003). Change in temperature is subsequently the dominant factor determining $CO_2$ concentration in brine channels (Geilfus et al., 2014). However, the incorporated sea ice algae take up $CO_2$ for photosynthesis and, together with other components of the microbial food web such as bacteria, release it during respiration. The balance of these processes commonly produces brine $CO_2$ concentrations that are lower in summer and higher in winter relative to atmospheric levels (Gleitz et al., 1996; Dieckmann and Thomas, 2003). Hare et al. (2013) examined the development of pH in an outdoor experimental sea ice facility in Winnipeg, Canada. Experiments were run over several weeks and ice attained a thickness of up to 22 cm. pH profiles had a characteristic 'C' profile with pH values greater than 9 at the surface and in the bottom 2 cm. Interior values were below that of the source water (8.4) with some as low as 7.1. The reason for this pattern was thought to be entirely due to abiotic factors as no algal growth was observed. The higher values at the surface and bottom reflected equilibrium with the atmosphere and underlying sea water, respectively, while those in the middle reflected $CO_2$ rejection during brine formation.

Changes in pH and $CO_2$ concentration in natural brines from the Weddell Sea in early summer have been documented by Papadimitriou et al. (2007). The *in situ* pH varied between 8.41 and 8.82, up more than 0.7 units from the underlying sea water. The concentration of $CO_2$ ranged from 3.1 µmol kg$^1$ to 15.9 µmol kg$^1$ and variations were consistent with biological activity (Papadimitriou et al., 2007).

As global temperatures warm, sea ice extent will inevitably decrease and the ice will form later in the season and melt sooner. As a consequence, average ice thickness will probably decrease. These large-scale changes are already clearly evident in the Arctic (Barnhart et al. 2016). Thinner ice would result in average irradiances within and beneath the ice being higher, although this will also be modified by changes in snow thickness. However, there are still large uncertainties in predictions of future changes in precipitation over sea ice and while a warmer atmosphere might be expected to increase snow fall, there is no clear evidence of this happening so far or whether in future it will fall as snow or rain (Bracegirdle et al., 2008; Leonard and Maksym, 2011).

## 3. Biological communities

Sea ice is not a solid, uniform, homogenous layer but instead contains many different habitats that can be colonized by a large range of organisms. Each of these habitats is characterized by different physical and chemical conditions. Initially any organism present in the underlying water column has the potential to be trapped within the ice as it forms. The early process of ice formation (frazil ice formation) can concentrate the phytoplankton biomass many times above that of the surrounding water column (Garrison et al., 1983). All species present in the water column can be harvested, although some, such as *Nitzschia stellata,* have the ability to greatly increase the likelihood of their capture by the release of ice active substances (Raymond et al., 1994; Ugalde et al., 2014). Young ice typically contains a highly diverse microalgal community (Scott et

al., 1994) but this diversity decreases with time as those species that are better adapted to this environment out compete those that are less well adapted. The biological communities can be differentiated by where they occur in the ice; in brine channels within the ice (brine communities), on the undersurface of the ice (bottom communities) or at the snow-ice interface (surface/infiltration communities) (Horner 1985; Arrigo, 2014; Bluhm et al. 2017) Bottom communities, which are dominant in both Arctic and Antarctic coastal locations, typically inhabit a benign and equitable environment characterized by minimal changes in temperature and salinity, maximum access to nutrients but low light levels. Brine communities, which dominate in pack ice, experience, extreme fluctuations in temperature and salinity, high to very low nutrient concentrations and intermediate light levels (Thomas and Dieckmann, 2003). Surface communities, which are often the dominant community in summer in pack ice, have access to the highest light but also experience the lowest temperature and nutrient levels. These communities also experience different $CO_2$/pH conditions. Bottom communities are constantly in contact with the underlying water column and typically experience the bulk ocean properties. Brine and surface communities, by contrast, experience supersaturated $CO_2$ concentrations during early ice formation but very low levels later in the season when the $CO_2$ is exhausted by photosynthesis. In coastal fast ice communities >90% of the algal biomass and productivity is concentrated in the bottom 10 cm. However, in pack ice communities the biomass is more evenly spread with significant biomass and productivity occurring both in brine channels and at the snow-ice interface (Meiners et al. 2012).

Major micro algal groups have been found to respond differently to $CO_2$ availability depending on physiological factors such as the presence of a carbon concentrating mechanism (CCM), which enables cells to use bicarbonate, or the type of RuBisCo present (Eberlein et al. 2014). Thus, the response of sea ice algal communities to ocean acidification will reflect their taxonomic composition. Diatoms are often the most abundant and also the most studied group of sea ice microbes, but other eukaryotic and prokaryote groups are often important. While diatoms tend to dominate bottom communities (Garrison, 1991), dinoflagellates and other flagellate groups tend to dominate interior and surface communities (Stoecker et al., 1992; Thomson et al., 2006). There is also a diverse bacterial community (Mock and Thomas, 2005; Bowman et al., 2012; Bowman, 2013, Bluhm et al., 2017) and evidence of an active microbial loop (Martin et al., 2012). While most diatoms, including sea ice species, have been found to possess a CCM, this has not been tested on most other groups and so their response to increase in $CO_2$ availability is unknown.

## 4. Ocean acidification experiments

### 4.1 Sea ice algae

Studies of ocean acidification effects on sea ice algae and bacteria have taken a number of different approaches. Some have just examined the effects of increased $CO_2$ concentration in isolation (Coad et al., 2016, Torstensson et al., 2015), while other have looked at the combined effects of co-stressors such as temperature (Pančić et al., 2015; Torstensson, 2012a,b), light (Xu et al., 2014a; Heiden et al., 2016) or iron (Xu et al., 2014b). Most studies have been undertaken on cultures of single species in a laboratory (Mitchell and Beardall, 1996; Xu et al., 2014a; Pančić et al., 2015; Torstensson et al., 2012a,b;

Young et al., 2015), although there has been one study of a brine community (Coad et al., 2016) and two *in situ* experiments on brine communities (McMinn et al, 2014; 2017). All but one of these studies (Torstensson et al., 2015) was conducted on a time scale of less than 30 days and most less than 10 days (Table 1).

Studied in isolation, increased $CO_2$ concentration seems only to have negative effects on growth and photosynthesis at levels above ~1000 µatm (Torstensson et al., 2015). When a natural diatom-dominated sea ice brine community was incubated over a $CO_2$ gradient from 400 µatm to 11,300 µatm for up to 18 days, only the treatment incubated at the most extreme concentration was significantly different from the control and other treatments (Coad et al., 2016). In a further experiment, they simulated future spring conditions by allowing a surface ice community to melt into seawater with the same large $CO_2$ gradient. Once again, only the most extreme $CO_2$ concentrations induced significant reductions in growth and photosynthesis. These results are consistent with most other studies of polar marine diatoms, which show increased growth rates and photosynthetic performance at $CO_2$ concentrations up to at least 1000 µatm (Trimborn et al., 2013; Hoppe et al., 2013). Future increases in marine $CO_2$ concentration, however, are likely to be accompanied by changes in temperature, light, macro nutrient and iron concentrations (Boyd et al., 2016) and interactions between these and other factors are likely to produce unpredictable results (Boyd and Hutchins, 2012). However, not all of these changes are likely to occur within the sea ice environment.

Studies of current Southern Ocean phytoplankton communities have identified the importance of iron and irradiance as being the key drivers determining changes in community composition and productivity (Feng et al., 2010). Others have suggested that future changes in $pCO_2$, temperature and stratification, which affects average irradiance and nutrient availability, will also be significant (Boyd et al., 2016; Constable et al., 2014). However, not all of these factors are directly relevant to ice algal communities. For instance, unlike the oceans, the temperature of sea ice habitats cannot rise above ~0°C. Increasing sea water temperatures may well change the temporal and spatial distribution of sea ice and ice thickness but they will have only a small effect on micro algal physiology.

The combined effects of temperature and $CO_2$ concentration on the bipolar sea ice/phytoplankton species, *Fragilariopsis cylindrus* was examined in short incubations (7 days) by Pančić et al. (2015). Significant interactions between temperature and pH were identified but the two factors produced opposite effects. Growth rates increased with increasing temperatures but decreased with decreasing pH, resulting in little overall change with treatment. As a result, it was concluded that *F. cylindrus* would largely be unaffected by increasing ocean acidification. These experiments were conducted over a wide range of temperature (1°C to 8°C) and pH (7.1 to 8.0). A similar study on the sea ice/benthic diatom *Navicula directa* likewise found only minor responses to increased $CO_2$ with no synergistic effects with temperature, although a smaller range of temperature (0.5- 4.5°C) and pH (7.9-8.2) was used (Torstensson et al., 2012a). These authors emphasized the need to examine multiple strains of a species and over longer incubation times before conclusions could be reached on its long-term response. It should be noted that while both these studies used sea ice algae taxa, the cells were exposed to temperatures well

above 0°C, i.e. temperatures that could never be experienced by cells living within sea ice. Although the maximum temperature of sea ice is fixed, the average temperature is still likely to rise. This could result in more intensive heterotrophic processes, via increased grazing rates and nutrient regeneration (Melnikov, 2009). However, it is probable that these processes, which already occur in spring and summer, will simply develop a little earlier.

Future changes in iron supply has been identified as likely to have a significant but unquantified effect on much of the world's phytoplankton (Shi et al., 2010) but the effects on sea ice communities are unknown; to date there have been no studies on the combined effects of increased $CO_2$ and iron-limitation on sea ice algae. A study of Southern Ocean phytoplankton, which contains many of the same species as sea ice communities, under co-limitation by $CO_2$, iron and other factors found that competition was likely to induce taxonomic changes in community composition, favoring small diatoms

(Xu et al., 2014b). Algae growing at the ice water interface (bottom communities), like the phytoplankton, experience iron limiting conditions and show evidence of chronic iron stress (Pankowski and McMinn, 2008; 2009). However, the iron concentration within brine channels is one to two orders of magnitude greater than in the underlying seawater (Lannuzel et al., 2011) and sea ice brine algae have been shown not to be iron stressed at all (Pankowski and McMinn, 2008; 2009). Under these circumstances, iron cannot be considered a co-stressor.

The combined effect of $CO_2$ concentration and increased light on the ice diatom, *Fragilariopsis curta* was examined by Heiden et al. (2016). In this study growth and photophysiology were not stimulated at relevant light (20 µmol photons m$^{-2}$ s$^{-1}$) and OA-relevant elevated $pCO_2$ (1000 µatm). These authors also showed that there were large variations in species-specific responses. The growth and photosynthetic response of the chlorophyte *Chlamydomonus,* isolated from Antarctic sea

ice, to ocean acidification and photoperiod was examined by Xu et al. (2014a) over the course of 28 days. While *Chlamydomonas* is not a common ice algal taxon, it none the less represents an organism adapted to living in the ice. Their data showed that $CO_2$ concentration had a minimal effect on the response to differing photoperiods.

There have been few *in situ* experiments with natural sea ice communities. McMinn et al. (2014, 2017) incubated

dinoflagellate-dominated brine algal communities *in situ* in McMurdo Sound, Antarctica. The 2014 study was conducted in spring while the 2017 study was conducted in late summer. They used sack holes to extract the brine (McMinn et al., 2009), adjusted the $CO_2$ concentrations of the treatments and then returned them to the same depth within the ice from where they were collected. Incubations were relatively short, up to 6 days. In both studies growth and photosynthesis were only affected when the pH fell below approximately 7.5. In a companion set of experiments in both studies the carbon system was

manipulated to hold the pH constant at ~8.0 while providing a gradient in $CO_2$ concentrations up to 2700 µatm. Growth increased by approximately 20% and remained constant at even the highest concentrations. This 20% increase in growth with increased $CO_2$ supply is of a similar scale to those estimated for a variety of temperate phytoplankton taxa on a similar $CO_2$ gradient (Rost et al., 2003; Riebesell, 2004).

The concentration of dissolved $CO_2$ in seawater is usually considered insufficient to maintain maximum growth rates in phytoplankton. As a consequence, most species have developed strategies to alleviate this stress; collectively referred to as Carbon Concentrating Mechanisms (CCM) (Raven et al., 2011). The presence of a CCM allows most marine microalgae, and diatoms in particular, to concentrate inorganic carbon as $CO_2$ and/or $HCO_3$ (Giordano et al., 2005) and thus alleviate much of this stress. Ocean acidification increases the natural $CO_2$ concentration and so potentially favours those species without a CCM, however, it will also reduce the need to invest in this energetically demanding function for species with CCMs (Beardall and Raven, 2004). Inorganic carbon uptake by the sea ice diatom *Nitzschia stellata* was examined by Mitchell and Beardall (1996). They demonstrated that this taxon was capable of carbon-dependent photosynthesis at rates greater than the $CO_2$ supply rate to the cell and that much of the carbonic anhydrase activity was associated with the cell surface; these characteristics demonstrated the presence of an active carbon concentrating mechanism (CCM). Other studies (Tortell et al., 2013; Gibson et al., 1999; McMinn et al., 1999) have also found evidence for CCM activity from either carbon isotope analysis or carbonic anhydrase activity. So far, all sea ice species examined have shown evidence of an active CCM. However, little is still known about how the presence or up/down regulation of CCMs will respond to future changes in $CO_2$ concentration (Raven et al., 2011).

While most sea ice algal OA research has focused on Antarctic communities, those from the Arctic would be expected to respond in a similar way as they share similar taxonomic compositions, i.e. dominated by diatoms, and have similar temperature, salinity and irradiance profiles (Horner 1985). There are, however, some structural differences in sea ice ecosystems between the two regions resulting in brine algal communities being virtually absent from the Arctic and the vast bulk of the algal biomass being located on the bottom of the ice (Horner, 1985, Thomas and Dieckmann, 2003). As in the Antarctic, bottom communities are generally less exposed to extreme fluctuations in temperature, salinity, gases and nutrients. Pančić et al. (2015) examined the response of the bipolar sea ice diatom species, *F. cylindrus*, isolated from the Greenland Sea, and concluded that, as in the Antarctic, OA would have little effect on the growth of this species.

## 4.2 Bacteria

Sea ice contains an abundant and diverse, psychrophilic bacterial community (Mock and Thomas, 2005; Bowman et al., 2012; Bowman, 2013). Ocean acidification effects on marine bacterial communities in general appear to be minimal (Oliver et al., 2014; Lin et al., 2017). They are able to mitigate the effect of decreased pH by enhancing the expression of genes encoding proton pumps, such as respiration complexes, proteorhodopsin and membrane transporters (Bunce et al., 2016). Bacterial growth rates usually appear to be closely coupled with phytoplankton growth (Engel et al. 2013). Likewise, in sea ice studies, bacterial growth rates have been found to increase with increasing $CO_2$ concentration, probably reflecting an increase in sea ice algae and an increased rate of DOC production (Torstensson et al., 2015). A pH range of 8.203 – 9.041 has also been found to have no effect on the community richness or diversity of Antarctic fast ice bacteria (Torstensson et al., 2013).

## 5. Discussion and Summary

Working with sea ice algal communities is implicitly more difficult than working with pelagic phytoplankton communities because, like benthic microbial ecosystems, they live in biofilms that have steep and highly structured vertical chemical and physical gradients. Most studies of sea ice algae physiology, including responses to pH and $CO_2$, have removed the cells from their ice substrate and suspended them in water. This process both exposes cells to osmotic shock and also destroys the highly structured vertical gradients. *In situ* studies, where the communities are not handled or moved, although more technically challenging, are likely to produce the most meaningful results. *In situ* methods have been used in studies of sea ice algal photosynthesis and photophysiology for several years (McMinn et al., 2000, 2012, Kuhl et al., 2001). An innovative *in situ* methodology for studying OA effects of on sea ice bottom communities has also been recently published by Barr et al. (2017), although no results from this study are yet forthcoming. This *in situ* methodological approach is similar to that used in 'Free Ocean Carbon Experiments' (FOCE) on Antarctic benthic communities (Stark et al., in press).

While studying sea ice bottom communities by *in situ* methods is achievable, studying brine communities *in situ* presents additional problems. These communities, which are entombed in brine channels and pockets, are often completely isolated and invisible. Currently used methods, such as variable fluorescence and microsensors, are unable to penetrate the ice matrix to measure biological responses. Even the *in situ* chemistry of the brine chambers cannot be measured. Studies by McMinn et al. (2012, 2014) extracted the brine algae from the ice by drainage into sack holes, manipulated the carbon chemistry and then reinserted the algal samples back into the ice. While not truly *in situ*, the algae in these studies were at least exposed to natural irradiances and temperatures. Future studies of the effects of OA on sea ice algal communities will also need to focus on using *in situ* approaches and more accurately reproducing natural conditions.

Most studies of the effects of OA on ice algae have used only short, i.e. less than 10 days, incubation times. In the first long term experiment (194 days), Torstensson et al. (2015) showed that significant impacts were not detected until after 147 days. Interestingly, however, most sea ice environments are both ephemeral and constantly changing. As pH and $CO_2$ concentrations within the ice are determined by temperature, natural sea ice communities are very unlikely to be exposed to a constant pH environment over this length of time. While longer experiments would seem to give more reliable responses, there is little point extending incubation times beyond the periodicity of natural change. Future studies could include these fluctuations into their experimental design.

It is well known that the physiological responses of diatoms in culture changes with time (Anderson, 2005). Some of the culture experiments discussed herein used strains of diatoms that had been isolated decades earlier (Xu et al. 2014, Heiden et al. 2016). Therefore, the extreme age of the culture used needs to be taken into account when interpreting the physiological response. Where possible, culture experiments should utilize freshly isolated strains to optimize meaningful physiological responses.

Brine algae and to a lesser degree bottom algae are naturally exposed to a large range of pH and $CO_2$ concentrations during seasonal ice formation and melting (Hare et al., 2013) and this range is typically greater than predicted changes resulting from OA. As a result, it is not surprising that ice algae appear to be well adapted to changes in carbon chemistry and show only minimal responses to these changes. Likewise, many coastal phytoplankton communities, which are also exposed to large diurnal and seasonal changes in pH and $CO_2$ concentrations, show either a positive response or no response to OA (Grear et al., 2017). Larger responses have been detected in open ocean areas where natural fluctuations are much lower (Li et al., 2016).

Sea ice ecosystems are ephemeral and in most relevant locations they melt and re-form each year. Thus, for some part of each year organisms inhabiting the ice must also survive outside of the ice, either as part of the phytoplankton or as resting spores on the bottom. During these times, they will be exposed to the full range of co-stressors that pelagic organisms experience (Xu et al., 2014c). Some sea ice taxa go into dormancy during these periods, e.g. *Polarella glacialis* (Thompson et al., 1996), while others, such as *Fragilariopsis curta* and *Fragilariosis cylindrus* make a major contribution to phytoplankton biomass. Most sea ice microorganisms do not appear to be badly impacted by moderate increases in $CO_2$. However, their ability to continue to make a major contribution to sea ice productivity will depend not only on their ability to survive in the ice but also on their ability to survive the increasing seawater temperatures, changing distribution of nutrients and declining pH forecast for the water column over the next centuries (Boyd and Hutchins, 2012).

**Competing Interests**

The author declares that he has no conflict of interest.

**Acknowledgements**

Australian Antarctic Science Grants program and Antarctic Gateway Partnership are acknowledged for their continuous financial support for sea ice research in general and for this review in particular.

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

**Figure Captions**

Figure 1. Generalised annual succession of pH and nutrient concentration in sea ice brine ecosystems in response to ice formation in winter and melting in spring. On the bar at the bottom of the figure, blue represents open water and white

5    represents the presence of ice. 'Nutrients' is a general trend showing the concentration of inorganic macronutrients.

| Species/Community | Days | T° | CO₂ (µatm) | Site | +/0/- | Author |
|---|---|---|---|---|---|---|
| | | | | | | |
| *Navicula directa* | 7 | 0.5-4.5° | 380-960 | Lab | 0 | Torstensson et al., 2012 |
| *Nitzschia lecointei* | 14 | -1.8°-2.5° | 390-960 | Lab | -/0 | Torstensson et al. 2013 |
| Brine community | 8 | -2.5° | 587-6066 | *In situ* | + | McMinn et al., 2014 |
| *Chlamydomonus* sp. | 28 | 5° | 390-1000 | Lab | 0 | Xu et al. 2014 |
| *Fragilariopsis cylindrus* | 7 | 1-8° | Ph 7.1-8.0 | Lab | 0 | Panic et al., 2015 |
| *Nitzschia lecointei* | 194 | -1.8° | 280-960 | Lab | 0/- | Torstensson et al., 2015 |
| Brine community | 8 | -5° | 400-11300 | lab | + | Coad et al., 2016 |
| *Fragilariopsis curta* | 4 | 4° | 180-1000 | Lab | + | Heiden et al., 2016 |
| Surface brine community | 6 | 0° | 45-4102 | *In situ* | + | McMinn et al., 2017 |

Table 1. Ocean acidification experiments on sea ice microorganisms. The column '+/0/-' indicates whether the organisms in the study responded positively '+', no change '0' or negatively '-' to the change in $CO_2$ concentration.

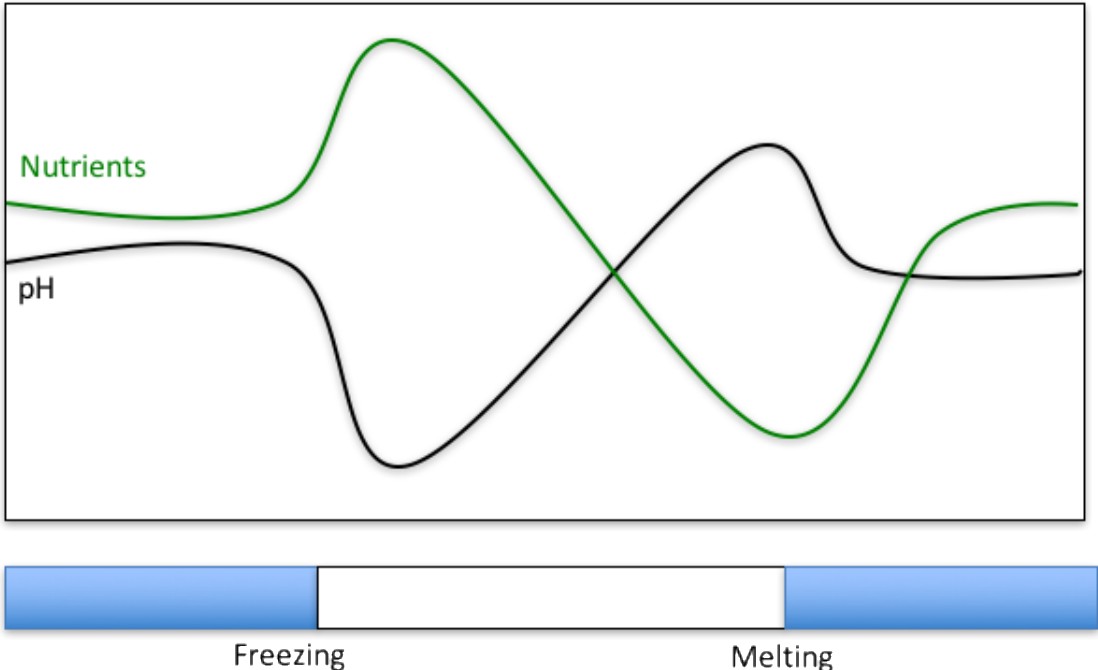

**Figure 1.**