# Peer review of "Reviews and syntheses: Ice Acidification, the effects of ocean acidification on sea ice microbial communities"

_Biogeosciences, 2017_

## Referee Comment (RC1) · Anonymous Referee #1 · 27 Apr 2017

Andrew McMinn has done a good job summarizing the (relatively scarce) literature that is available about effects of ocean acidification on sea ice microorganisms. I think this is an important topic, and I am generally in favor of the publication of this type of synthesis paper. However, I do believe that the author needs to elaborate the review paper before it is ready for publication. I feel like the manuscript summarizes the main findings from the studies quite well, but I am lacking key points and overall conclusions from the synthesis of these papers. I would like to see more emphasis on general conclusions that can be drawn from reviewing the literature, other than a collection of findings (albeit solid and worth publishing). More specifically, my key points are:

I am lacking a section about future directions on the topic (and perhaps an overall summary in the end of the review). What are the key points lacking in the field of sea ice acidification and how should we best approach them? For instance, the author dis-

cusses strong pH fluctuations in sea ice – how can we best address these fluctuations experimentally?

I would also like to see that the author comments on the technical issues working with ocean acidification in sea ice – is there a "best practice" approach for doing it? The author lists a number of approaches (e.g. cultures, brine communities, in situ experiments). Ocean acidification experiments are relatively complicated to perform by themselves, considering the effort needed to control the $CO_2$-system well. Doing it in sea ice may be intimidating for scientists new to sea ice work. I believe that suggestions for a best practice approach would be very helpful to our community to increase the scientific effort on the topic (especially in the Arctic).

Why do you think everything is performed in the Southern Ocean? Would you expect to see differences in the Arctic, considering the differences in sea ice characteristics?

I would also be happy to hear about the author's opinion about the importance of ice acidification in relation to planktonic acidification. By the limited number of published studies available on ice algae, can you draw any conclusions about the sensitivity of sea ice algae compared to phytoplankton? Is ocean acidification more important in the water mass than in sea ice?

I am also a bit reluctant about Figure 1. It describes a highly generalized seasonal cycle of pH and nutrients. Why are nutrients there, and what type of nutrients is the author referring to (I am assuming inorganic macronutrients)? Nutrients are never discussed in relation to Figure 1 in the text, so please elaborate on this. Some nutrients (mainly phosphourous) are accumulated in the brine before the melt in summer (Fripiat et al 2017, Elementa), so I think it is important to note that this figure is quite generalized.

I would also like to bring some additional papers to the author's attention, which are not discussed in the review but could perhaps be relevant for the discussion. The main papers that I am referring to are Barr et al (2017, Limnology and oceanography: Methods), Søgaard et al (2011, Polar Biology) and Torstensson et al (2013, Biogeosciences).

Minor points:

Title: Please revise the title so that it starts with "Reviews and synthesis:", according to the journal's instructions.

P3, L31: Please clarify "ice water interface (surface communities)". This term might be confusing for a reader who lacks knowledge about surface flooding (if that is what the author is referring to). I believe that many readers could misinterpret the "ice-water interface" as the bottom community.

P4, L7-11: This section needs some elaboration. Please explain how it relates to ocean acidification studies and the context of this review.

P5, L6-8: Please elaborate this statement. I would imagine that temperature would affect sea ice thickness, melt pond formation and less multiyear ice (in the Arctic at least), which definitely have large impacts on the environment in sea ice.

P7, L6-12: What about indirect physiological effects on bacterial communities from changes in algal physiology? Engel et al (2013, Biogeosciences) reported elevated bacterioplankton growth due to shifted nutrient stoichiometry and DOC production in Arctic phytoplankton.

Table 1: Would a column with a short summary of the results (e.g. positive/negative/no effect) be in place in this table? That way this could be a central Table summarizing the whole review quite well.

---

## Referee Comment (RC2) · Anonymous Referee #1 · 27 Apr 2017

Additional comment:

I would like the author to carefully check the bibliography. There are a number of papers missing in the reference list, that are listed in the text and vice versa. For instance, Søgaard et al (2011) is listed in the bibliography, but never in the text. Young et al (2015) and Ugalde et al (2014) are referred to in the text, but it is missing in the bibliography, as far as I can see. There might be more instances, so please make sure that the referencing is in order.

---

## Referee Comment (RC3) · Anonymous Referee #2 · 18 May 2017

General comment Ocean acidification (OA) is a hot topic that received increasing attention during the last 10 years or so. Several experimental studies have been conducted to assess the sensitivity of phytoplankton and other marine organisms to the predicted changes in pH/CO2 concentrations. Bacteria and microalgae living in the sea ice will also be exposed to changes in pH. Few studies so far attended to determine the sensitivity of these microorganisms to changes in pH/CO2. This paper review their main findings. The paper reads well and provides a good overview of our state of knowledge. Although bacteria are also considered here, the focus is clearly on sea ice algae. The paper offers a good balance between the factual review of the findings from the different papers and more personal viewpoints. Surprisingly for such a specific and relatively recent topic, the number of published papers is large enough to justify a review. The main conclusion is that ice-related organisms are generally quite resistant

to OA although the potential co-effect of additional stressors such as iron limitation is uncertain.

I would recommand to:

1. Provide an estimate of the relative importance of bottom, brines, and top ice assemblages to the annual primary productivity. In the present version of the paper, algae living in these different ice 'habitats' seem to be equally important in terms of PP which is not the case.

2. Present the information on bacteria and algae in different sections.

Specific comments

P1, 10: ...than marine phytoplankton...Note that coastal and even more estuarine phytoplankton are also subjected to large variations in pH taking place at different time scales.

P1, 23: ...on bacterial growth...

P2, 20-21: Is this seasonality found all over the SO or only in the marginal ice zone? The 2-3 examples provided in the paragraph are all from the near coastal waters.

P2, 32: ...CO2 concentration in....?

P3, 14-19: This paragraph disrupts the flow of the paper.

P3, 14: ...form later in the season and melt sooner...Yes but the extent of sea ice tends to decrease over most the Antarctica waters.

P3, 30: ...The biological communities can be...Here the author should refer to the previous studies describing these different assemblages (ex. Cota et al. 1991, Horner et al. 1992, and the more recent one by Bluhm et al. 2017). In Antarctica ice, infiltration assemblage are important at time. They are not mentioned in the review.

P4, 9: ...communities (add Bluhm et al. 2017 in the list)...

P4, 28: . . .diatoms, which also show increased. . .I am not sure 'also' fits well here since there are several mentions before (and after. . .) of no or negative effects of high CO2 on phytoplankton growth. This could be confusing.

P5, 4: . . .which affects average. . .

P5, 32: Delete 'at all'.

P6, 1: . . .the important. . .delete 'important' or explain why this species is important.

P6, 2: . . .Unlike most previous experiments, growth was not stimulated. . .Why 'unlike'? You previously mentioned other studies showing no effect of high CO2 on phytoplankton growth.

P6, 9-12: This paragraph will better fit at the end in a 'summary' section.

P6, 30: . . .demanding function for species with CCMs. . .

P7, 3: . . .how these processes. . .Which processes? Please be more specific.

P7, 10: . . .Likewise. . .The author should explain why bacterial growth increase with increasing CO2 concentrations.

P7, 14: . . .Sea ice ecosystems. . .The idea developed in this paragraph is interesting but is not well introduced. The paper needs a proper conclusion section starting with a short summary of the main findings, followed by the limitations identified (ex. age of the culture), and ending perhaps with the importance of considering the full life cycle of the species.

---

## Author Comment (AC1) · 3 Jul 2017

Author's final comments

I agree with all comments made by RC1 and have followed all suggestions. Of the larger changes, I have;

1. included a final section '5. Discussion and summary', where I have added most of the requested additional discussion.

2. Changed the Table to include extra studies and a column indicating positive/negative or no change

3. Figure 1, I have added to the caption to better explain the role of nutrients

[Figure]

Anonymous Referee #1

Andrew McMinn has done a good job summarizing the (relatively scarce) literature that is available about effects of ocean acidification on sea ice microorganisms. I think this is an important topic, and I am generally in favor of the publication of this type of synthesis paper. However, I do believe that the author needs to elaborate the review paper before it is ready for publication. I feel like the manuscript summarizes the main findings from the studies quite well, but I am lacking key points and overall conclusions from the synthesis of these papers. I would like to see more emphasis on general conclusions that can be drawn from reviewing the literature, other than a collection of findings (albeit solid and worth publishing). More specifically, my key points are:

I am lacking a section about future directions on the topic (and perhaps an overall summary in the end of the review). What are the key points lacking in the field of sea ice acidification and how should we best approach them? For instance, the author discusses strong pH fluctuations in sea ice – how can we best address these fluctuations experimentally?

I would also like to see that the author comments on the technical issues working with ocean acidification in sea ice – is there a "best practice" approach for doing it? The author lists a number of approaches (e.g. cultures, brine communities, in situ experiments). Ocean acidification experiments are relatively complicated to perform by themselves, considering the effort needed to control the $CO_2$-system well. Doing it in sea ice may be intimidating for scientists new to sea ice work. I believe that suggestions for a best practice approach would be very helpful to our community to increase the scientific effort on the topic (especially in the Arctic).

RESPONSE: I agree with all the above comments and have now added an extra section '5. Discussion and Summary' at the end of the manuscript where I address all these points. In this section I discuss future directions, technical issues and best practice, and a summary

Why do you think everything is performed in the Southern Ocean? Would you expect to see differences in the Arctic, considering the differences in sea ice characteristics? I would also be happy to hear about the author's opinion about the importance of ice acidification in relation to planktonic acidification. By the limited number of published studies available on ice algae, can you draw any conclusions about the sensitivity of sea ice algae compared to phytoplankton? Is ocean acidification more important in the water mass than in sea ice?

RESPONSE: There are currently no published studies based on the Arctic. However, I have a added a sentence to indicate Arctic bottom communities are expected to respond in a similar manner to Antarctic communities.

I am also a bit reluctant about Figure 1. It describes a highly generalized seasonal cycle of pH and nutrients. Why are nutrients there, and what type of nutrients is the author referring to (I am assuming inorganic macronutrients)? Nutrients are never discussed in relation to Figure 1 in the text, so please elaborate on this. Some nutrients (mainly phosphourous) are accumulated in the brine before the melt in summer (Fripiat et al 2017, Elementa), so I think it is important to note that this figure is quite generalised.

RESPONSE: I like this diagram and want to leave it in. I accept the reviewer's comments, however, and have expanded the caption to better explain its meaning. I have also added a sentence to the introduction describing the expulsion of nutrients during ice formation (p2, ln 16).

I would also like to bring some additional papers to the author's attention, which are not discussed in the review but could perhaps be relevant for the discussion. The main papers that I am referring to are Barr et al (2017, Limnology and oceanography: Methods), Søgaard et al (2011, Polar Biology) and Torstensson et al (2013, Biogeosciences).

RESPONSE: Added Barr et al. 2017 and Torstensson et al. 2013 to refs, Torstensson et al. 2013 is now also in Table 1. I haven't added Sogaard et al. 2013 as they only looks at elevated pH, i.e ocean alkalination not acidification.

Minor points:

Title: Please revise the title so that it starts with "Reviews and synthesis:", according to the journal's instructions.

RESPONSE: Changed as suggested

P3, L31: Please clarify "ice water interface (surface communities)". This term might be confusing for a reader who lacks knowledge about surface flooding (if that is what the author is referring to). I believe that many readers could misinterpret the "ice-water interface" as the bottom community.

RESPONSE: Changed to snow-ice interface

P4, L7-11: This section needs some elaboration. Please explain how it relates to ocean acidification studies and the context of this review.

RESPONSE: I have added some extra context here, emphasizing that different groups have differing physiological responses depending on whether they have a CCM or the type of RuBisCo present (p4, Ln 15-24)

P5, L6-8: Please elaborate this statement. I would imagine that temperature would affect sea ice thickness, melt pond formation and less multiyear ice (in the Arctic at least), which definitely have large impacts on the environment in sea ice.

RESPONSE: I am trying to focus on physiological responses here. Phytoplankton will need to respond to increased temperatures, sea ice algae will not. I have restructured this paragraph to make this clearer. I agree that there will be impacts on the physical and chemical structures.

P7, L6-12: What about indirect physiological effects on bacterial communities from changes in algal physiology? Engel et al (2013, Biogeosciences) reported elevated bacterioplankton growth due to shifted nutrient stoichiometry and DOC production in Arctic phytoplankton.

RESPONSE: I have added some general comments about bacteria and phytoplankton but there is very little specific information about sea ice. I have also added the Engel reference, although it relates to polar phytoplankton. Table 1: Would a column with a short summary of the results (e.g. positive/negative/no effect) be in place in this table? That way this could be a central Table summarizing the whole review quite well.

RESPONSE: I have added the extra column ad included a more detail Table caption

---

## Author Comment (AC2) · 3 Jul 2017

Anonymous Referee #1: Additional comment:

I would like the author to carefully check the bibliography. There are a number of papers missing in the reference list, that are listed in the text and vice versa. For instance, Søgaard et al (2011) is listed in the bibliography, but never in the text. Young et al (2015) and Ugalde et al (2014) are referred to in the text, but it is missing in the bibliography, as far as I can see. There might be more instances, so please make sure that the referencing is in order.

RESPONSE: I have rechecked the reference list to include all cited publications and exclude those that aren't. In particular; Young and Ugalde have been added to references. I have removed Søgaard et al (2011), who only discuss elevated pH.

---

## Author Comment (AC3) · 3 Jul 2017

I thank the review for his/her constructive criticisms. I agree with everything suggested and have made changes accordingly.

Anonymous Referee #2

General comment Ocean acidification (OA) is a hot topic that received increasing attention during the last 10 years or so. Several experimental studies have been conducted to assess the sensitivity of phytoplankton and other marine organisms to the predicted changes in pH/CO2 concentrations. Bacteria and microalgae living in the sea ice will also be exposed to changes in pH. Few studies so far attended to determine the sensitivity of these microorganisms to changes in pH/CO2. This paper reviews their main

findings. The paper reads well and provides a good overview of our state of knowledge. Although bacteria are also considered here, the focus is clearly on sea ice algae. The paper offers a good balance between the factual review of the findings from the different papers and more personal viewpoints. Surprisingly for such specific and relatively recent topic, the number of published papers is large enough to justify a re- view. The main conclusion is that ice-related organisms are generally quite resistant

I would recommend to:

Provide an estimate of the relative importance of bottom, brines, and top ice assemblages to the annual primary productivity. In the present version of the paper, algae living in these different ice 'habitats' seem to be equally important in terms of PP which is not the case.

RESPONSE: I have modified this section to explain that each of these communities can be dominant in different seasons and different locations.

Present the information on bacteria and algae in different sections.

RESPONSE: Now broken into different sections, as suggested

Specific comments

P1, 10: . . .than marine phytoplankton. . .Note that coastal and even more estuarine phytoplankton are also subjected to large variations in pH taking place at different time scales.

RESPONSE: RESPONSE: Yes, I have added 'like some coastal and estuarine phytoplankton'

P1, 23: . . .on bacterial growth. . .

RESPONSE: Added 'bacterial'

P2, 20-21: Is this seasonality found all over the SO or only in the marginal ice zone?

The 2-3 examples provided in the paragraph are all from the near coastal waters.

RESPONSE: Added 'in the seasonally ice-covered waters'

P2, 32: . . .CO2 concentration in. . ..?

RESPONSE: Added 'in brine channels

P3, 14-19: This paragraph disrupts the flow of the paper.

RESPONSE: I have now removed this paragraph to the final section

P3, 14: . . .form later in the season and melt sooner. . .Yes but the extent of sea ice tends to decrease over most the Antarctica waters.

RESPONSE: Changed to 'sea ice extent will inevitably decrease and the ice will form later in the season and melt sooner'.

P3, 30: . . .The biological communities can be. . .Here the author should refer to the previous studies describing these different assemblages (ex. Cota et al. 1991, Horner et al. 1992, and the more recent one by Bluhm et al. 2017). In Antarctica ice, infiltration assemblages are important at time. They are not mentioned in the review.

RESPONSE: Added Horner 1992, Bluhm et al 2017, Arrigo 2014. Infiltration communities are a type of surface community, which are discussed here extensively. Recent reviews (eg Arrigo 2014) refer to infiltration communities to as 'surface communities' and do not specifically emphasize the mechanism of their formation. I have added 'infiltration to the first time I comment on surface communities in both the abstract and the section on biological communities (p1 ln 15; p4 ln 4).

P4, 9: ...communities (add Bluhm et al. 2017 in the list. . .

RESPONSE: Added Bluhm et al. 2017

P4, 28: . . .diatoms, which also show increased. . .I am not sure 'also' fits well here since there are several mentions before (and after. . .) of no or negative effects of high

CO2 on phytoplankton growth. This could be confusing.

RESPONSE: Deleted 'also'

P5, 4: . . .which affects average. . .

RESPONSE: Changed to 'affects'

P5, 32: Delete 'at all'.

RESPONSE: Lannuzel et al. 2011 is correct

P6, 1: . . .the important. . .delete 'important' or explain why this species is important.

RESPONSE: Deleted

P6, 2: . . . .Unlike most previous experiments, growth was not stimulated. . .Why 'unlike'? You previously mentioned other studies showing no effect of high CO2 on phyplank- ton growth.

RESPONSE: Deleted 'unlike most previous studies'

P6, 9-12: This paragraph will better fit at the end in a 'summary' section. P6, 30: . . .demanding function for species with CCMs. . .

RESPONSE: I have moved to the final paragraph

P7, 3: . . .how these processes. . .Which processes? Please be more specific.

RESPONSE: Added 'the presence or up/down regulation of CCMs'

P7, 10: . . .Likewise. . .The author should explain why bacterial growth increase with increasing CO2 concentrations.

RESPONSE: Added 'because of increased DOC production'

P7, 14: . . .Sea ice ecosystems. . .The idea developed in this paragraph is interesting but is not well introduced. The paper needs a proper conclusion section starting with

a short summary of the main findings, followed by the limitations identified (ex. age of the culture), and ending perhaps with the importance of considering the full life cycle of the species.

RESPONSE: I have added a new section '5. Discussion and summary' This section expands on the ideas suggested here and adds a final summary.